# Social implications of the 30×30 global conservation target

Javier Fajardo [1,2] ✉, Heather C. Bingham [3], Dan Brockington [2,4,5], Rebecca Chaplin-Kramer [6], James A. Fitzsimons [7,8,9], Forrest Fleischman [10], Alain Frechette [11], Rachael D. Garrett [1], Carolina Hazin [12], Tobias Kuemmerle [13,14], Janeth Lessmann [3], Milagre O. F. Nuvunga [15], Brian O'Donnell [16], Fred Onyai [17], Ruth Pinto [18], Marion Pfeifer [19], Rose Pritchard [20], Casey M. Ryan [21], Priya Shyamsundar [22], Josefa Cariño Tauli [23,24], David Mwesigye Tumusiime [25], Jasmin Upton [3], Gary R. Watmough [21], Julie G. Zaehringer [26,27] & Chris Sandbrook [1] ✉

Target 3 of the Kunming-Montreal Global Biodiversity Framework aims to increase global protected and conserved area coverage to at least 30% by 2030. The impact on people, whether positive or negative, will depend on the social context of additional areas and how they are governed and managed. Here, we show that Target 3 could affect large and socially diverse populations under different implementation scenarios. Nearly half the human population lives within 10 km of areas included in a scenario maximising biodiversity representation. Four percent live near areas included in an Indigenous and traditional territories-based scenario, including many in areas with low Human Development Index scores (74%) and high participation in wild harvesting (91%). A scenario prioritising nature's contributions to people is intermediate on all measures. Our results demonstrate that Target 3 is a highly ambitious social as well as ecological target, requiring an equally ambitious commitment to development funding and support for local residents.

Protected and conserved areas are crucial for achieving conservation goals[1]. In 2022, Parties to the Convention on Biological Diversity adopted the Kunming-Montreal Global Biodiversity Framework (KMGBF), including Target 3 (known as the 30 × 30 target), which aims to:

Ensure and enable that by 2030 at least 30 per cent of terrestrial and inland water areas, and of coastal and marine areas, especially areas of particular importance for biodiversity and ecosystem functions and services, are effectively conserved and managed through ecologically representative, well-connected and equitably governed systems of protected areas and other effective area-based conservation measures, recognizing Indigenous and traditional territories, where applicable, and integrated into wider landscapes, seascapes and the ocean,

while ensuring that any sustainable use, where appropriate in such areas, is fully consistent with conservation outcomes, recognizing and respecting the rights of Indigenous peoples and local communities, including over their traditional territories[2].

Here, we use the term protected and conserved areas to refer to all areas contributing towards the spatial element of Target 3 through three approaches: Protected Areas (PAs), Other Effective Area-based Conservation Measures (OECMs), and, where applicable, Indigenous and Traditional Territories (ITTs)[3,4]. Whilst PAs are primarily managed for conservation, OECMs recognise other areas that make important long-term contributions to conservation regardless of their management objectives[5]. ITTs are areas governed and conserved by Indigenous Peoples and custodian communities, though there is currently no agreed definition and processes are ongoing to define ITTs within the

---

context of the CBD. ITTs can be recognised within a framework of PAs and OECMs, where freely chosen by custodian communities, or potentially they could be recognised as contributing to Target 3 on their own terms as ITTs[6,7]. However, how ITTs could be recognised and incorporated into Target 3 remains contested and is likely to vary within and between countries[3,8]. In all cases, it is necessary to respect free, prior and informed consent and ensure appropriate recognition— that is, recognized and respected in accordance with custodians' wishes, rights, and institutions.

In August 2024, the combined extent of reported PAs and OECMs was 17.5% of terrestrial areas and inland waters[9]. This figure needs to nearly double by 2030 to meet the spatial coverage element of Target 3 on land, all while complying with other elements of the target, including ecological representation, connectivity, upholding rights of Indigenous Peoples and local communities, and equitable governance[10,11]. Currently, PAs and OECMs under the governance of non-state actors are under-recognised and under-reported[12,13]. The appropriate recognition of ITTs could be key to achieving Target 3, given their crucial role in protecting biodiversity and in environmental governance. For example, estimates suggest that land owned or governed by Indigenous Peoples and local communities covers at least 32% of Earth's terrestrial surface area, and likely more due to a lack of data and under reporting[14].

While there is widespread support for the 30 × 30 target by residents in countries surveyed[15,16], protected and conserved areas can have diverse positive and negative impacts on people, making it crucial to understand the social implications of expanding their coverage[17–19]. PAs can bring benefits locally, by supporting livelihoods and protecting cultural and spiritual sites[20–23] and at a broader scale, such as through climate regulation and nutrient cycling[24–26]. However, PAs can also bring costs, particularly locally and to groups in marginalised and vulnerable situations (often with greater negative impacts on women and girls[23,27]), due to displacement, restriction and criminalisation of access to traditional lands and resources, and increased human-wildlife conflict[28,29]. Social impacts of OECMs are less well studied[30], but could cover a similar range to PAs. ITTs tend to have positive impacts on residents' well-being, depending on the quality of governance and tenure security[31].

The social impacts of Target 3 will depend on how and where it is implemented. The KMGBF text leaves both these questions open to interpretation by parties and other implementing actors, all of whom have their own values and priorities[19,32]. On the "how" question, protected and conserved areas vary substantially in their approach to management (from strict protection restricting residence and use of natural resources through to permanent residence and sustainable use) and governance (e.g., by governments, private actors, Indigenous Peoples, local communities or combinations of these[33–35]). There is evidence that more inclusive and locally-led area-based conservation can lead to better outcomes for people and biodiversity[36–39]. These arrangements can apply to certain forms of PA and OECM, but must apply to ITTs which are governed through traditional institutions and supported through the recognition of rights and tenure[31].

On the "where" question, site location will determine the number and circumstances of people who will be affected by the establishment or recognition of protected and conserved areas. Understanding the social context of those likely to be affected is crucial for any successful implementation of Target 3, as this information is required for good decision-making about management, governance and allocation of resources. However, little is known at present about these social conditions at the global scale, beyond broad estimates of numbers of resident people[40–42]. This is in contrast to assessments of landscape restoration, which have highlighted the risk of interventions that discount local needs, particularly for people with among the lowest incomes, education levels and health outcomes[38,43].

Here, we conduct a global assessment of the number and socio-economic conditions of people living in and close to terrestrial areas and inland waters that could be recognised as protected and conserved areas for their potential contribution to achieving Target 3. We do so using three spatial scenarios for achieving the 30% coverage on land and inland waters element of Target 3, based on published studies, each reflecting different priorities for site selection. These are: (1) a biodiversity-based approach that expands the protected and conserved area network in a way that maximises biodiversity representation[42]; (2) a Nature's Contributions to People (NCP)-based approach that maximises representation of areas that contribute local and global services to people[44]; and (3) an Indigenous and traditional territories approach that prioritises ancestral lands governed by Indigenous Peoples and custodian communities with high biodiversity value[45] (Fig. 1). Each scenario builds on the current coverage by PAs and OECMs, adding complementary areas selected by following the priorities of the scenario to reach 30% terrestrial coverage. We perform spatial analyses to assess the overlap between areas included in each scenario (with and without a 10 km buffer) and global gridded datasets for several critical social variables: population[46], Human Development Index[47] (HDI), wild harvesting[48] (in tropical areas), smallholder farming[49] and livestock rangelands[50]. We include the 10 km buffer as people often experience effects, whether positive or negative, from nearby protected and conserved areas[51–53]. To capture regional variability and address the inherent uncertainties in social datasets, we also conduct analyses at the continental scale and compare alternative datasets for population size, development status, and livelihoods to validate our findings and identify potential discrepancies.

We find that the number and socioeconomic characteristics of people living in or near additional conservation areas differ markedly across scenarios. These contrasting profiles demonstrate that alternative approaches to achieving Target 3 bring different social contexts for implementation, each with its own challenges and opportunities. These findings underscore the need for context-sensitive approaches, adequate funding and careful integration of social considerations to ensure that implementation is equitable and effective. We do not attempt to predict the final social outcomes of implementation of the scenarios, as these will depend on how sites are governed and managed, which in turn reflect political choices about how countries choose to meet the target within the limited time available.

## Results
### Spatial distribution of scenarios
Target 3 is a global, rather than national, target, and the contribution of different countries and regions is likely to be highly variable[42]. This variability is reflected in the spatial differences among the three scenarios considered, as illustrated by the Amazonia region (Fig. 1). The Biodiversity scenario has a much wider distribution of relatively small areas (partly reflecting hotspots of species endemism), whereas the other scenarios are more aggregated, particularly in tropical forest regions for the NCP scenario (reflecting the large contribution to critical services these ecosystems are currently assessed as providing). This difference in spatial configuration also influences the extent of 10 km buffer zones, with scenarios featuring smaller, dispersed areas having more extensive buffer zones compared to those with more aggregated areas (Supplementary Fig. 1). Overlap between scenarios is limited (Supplementary Figs. 2 and 3). The ITT scenario is the most distinct, with 76% of its area unique to it, compared to approximately 60% in the other scenarios. The Biodiversity and NCP scenarios have 24% overlap, while their overlap with the ITT scenario is 8 and 12%, respectively. Only 4.7% of new areas are shared among all three scenarios, covering just 0.6% of the global terrestrial and inland water area. Further, 51% of Earth's terrestrial and inland water areas are not included in any of the scenarios.

## Target 3 scenarios

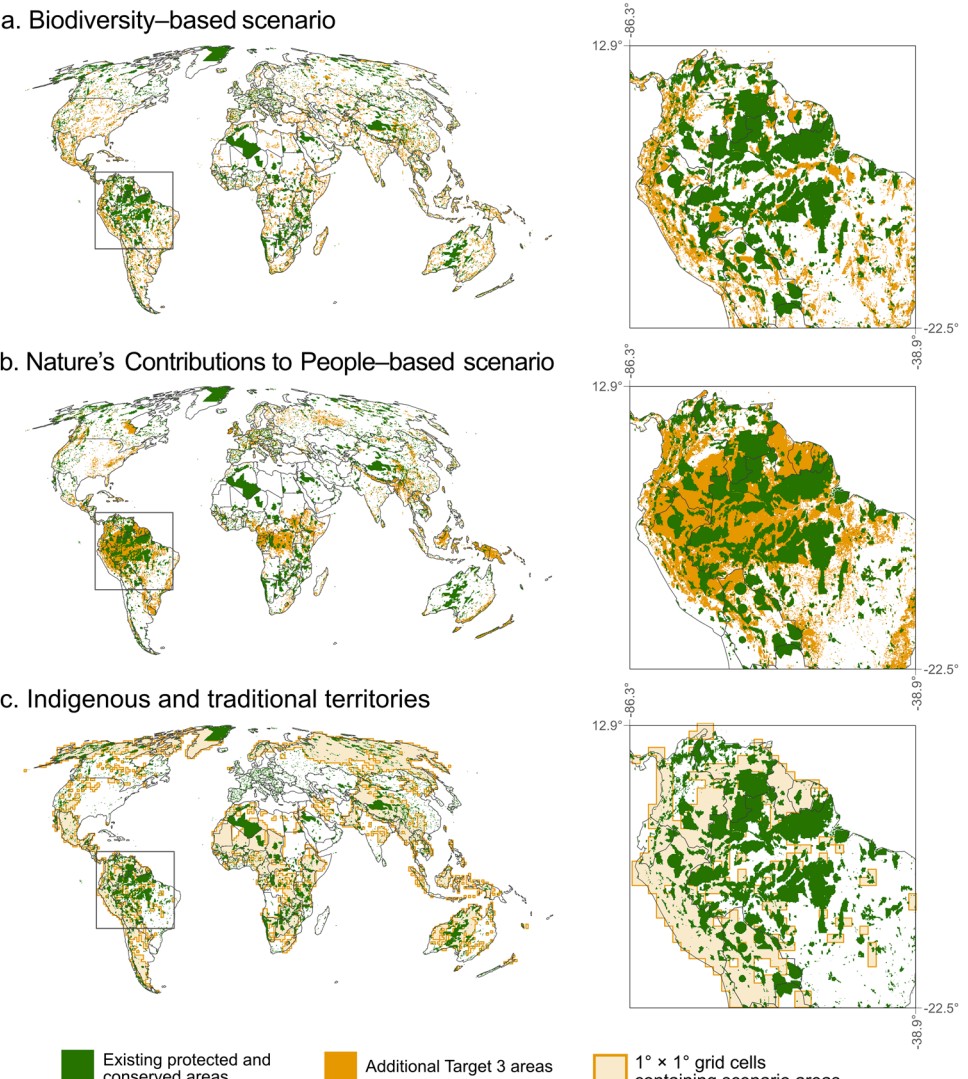

**Fig. 1 | Maps of Target 3 scenarios.** Additional global terrestrial and inland water areas that could be conserved to meet Target 3 scenarios are presented in orange for the **a** Biodiversity-based scenario, **b** Nature's Contributions to People (NCP)-based scenario and **c** Indigenous and traditional territories (ITT) scenario. Existing protected and conserved areas are shown in green. For the ITT scenario, orange shading represents 1° × 1° grid cells in which the scenario is present, rather than the precise spatial extent of the scenario area. This coarser representation is used to avoid displaying sensitive boundaries related to Indigenous and traditional territories and areas conserved by Indigenous peoples and custodian communities, and therefore visually overestimates the coverage.

### Human population and development status

Approximately 396 million people currently reside within existing PAs and OECMs, with a further 1.15 billion living within 10 km of these areas (Fig. 2A, see expanded results in Supplementary Table 1). Two of our three scenarios would substantially increase the number of people living in or close to protected and conserved areas. The Biodiversity scenario would result in the largest increase, with the resident population reaching 2.2 billion, or 29.9% of the global population—more than five times the current number despite less than a doubling of area. The population within 10 km of a protected or conserved area would also rise substantially, reaching 2.7 billion people. The NCP scenario would increase the resident population within protected and conserved areas to one billion (13.5% of the global population) with 2.3 billion within 10 km. The lower population in the NCP scenario compared to the Biodiversity scenario reflects its spatial configuration with some larger, contiguous areas in regions with low population density (Fig. 1), also resulting in less extensive buffer zones (Supplementary Fig. 1). The ITT scenario would increase the resident population to 517 million (6.8% of the global population), with 1.3 billion within 10 km. The populations residing in overlap areas among scenarios show no substantial deviations from these patterns, with population sizes and densities broadly reflecting intermediate conditions between the scenarios they connect (Supplementary Fig. 4). These results are robust to estimation with three alternative population datasets (see "Sensitivity analysis" Supplementary Fig. 7). They are also consistent when scenario layers are modified, including an alternative national-level prioritisation of NCPs (Chaplin-Kramer et al. 2022) and an alternative ITT scenario covering 36% of terrestrial areas to be more inclusive of all globally mapped ITTs (Supplementary Fig. 5; see "Sensitivity analysis" Supplementary Fig. 8). These numbers are likely to increase with time, particularly as the spatial overlaps of wildlife and people are expected to grow due to increase in human population densities[54]. Projected future population growth to 2030 within the Biodiversity and NCP scenarios are around 11%, slightly above the global rate of 9% (Supplementary Fig. 6). Population growth in the ITT scenario area by 2030

**Fig. 2 | Population and socioeconomic characteristics of areas under existing protected and conserved areas and additional areas under Target 3 scenarios. a** Total resident and neighbouring population of current protected and conserved areas and within a 10 km buffer of those areas, and additional numbers for areas under expansion scenarios and within a 10 km buffer of those areas. **b** Human Development Index (HDI) status of resident population of expansion areas, expressed as percentage of the overlapped population in different HDI categories. **c** Livelihoods of resident population of expansion areas, expressed as the percentage of population by wild harvesting status of households (tropical population), farm area, disaggregated by small farms (≤ 5 ha), large farms (> 5 ha) and non-farmed area, and livestock rangeland area. Low HDI: <0.550; Medium HDI: 0.550–0.699; High HDI: 0.700–0.799; Very High HDI: >0.800.

is nearly 30% (over three times the global rate) but from a much lower starting population.

There are striking differences among scenarios in the development status (Fig. 2B and Supplementary Table 2). Compared to those in current protected and conserved areas, the population in new areas in the Biodiversity scenario has a lower proportion of people in the low HDI category (10.8% vs 16.2%; HDI < 0.55; Fig. 2B) but nearly double the proportion of people in the medium HDI category (33.9% vs 17.5%; HDI: 0.55–0.69). The population affected by the NCP scenario has broadly similar HDI proportions to current protected and conserved areas, but with a greater proportion in the medium HDI category (30.5%). The ITT scenario is very different, with 61.0% of the affected population in the low HDI category and over 90% in the low and medium HDI categories combined. However, the absolute number of people in the low HDI category is greatest in the Biodiversity scenario (195 million people, which is 21.4% of all people globally with low HDI), due to its larger total population, compared to 62 million in currently protected and conserved areas (6.8%) and 118 million (12.9%) and 73 million (8.0%) in the NCP and ITT scenarios, respectively. Estimations using alternative subglobal well-being metrics produce consistent results (Supplementary Fig. 9).

Implementing Target 3 could also affect the livelihoods of resident populations[55]. The livelihood profiles of the populations living within potential new protected and conserved areas varied across scenarios (Fig. 2C and Supplementary Table 3). Compared to those in existing protected and conserved areas, the Biodiversity scenario has a lower proportion of people living in the tropics who use wild harvesting (34.1% Biodiversity scenario vs 55.6% in existing protected and conserved areas) and of livestock rangeland area (33.4% vs 47.3%), and a higher proportion of farmland (60.8% vs 29.8% of the scenario area). The NCP scenario has a slightly higher proportion of people who use wild harvesting (66.7%) and area of farmland (54.9%), but a smaller proportion of livestock area (19.3%), whereas the ITT scenario has a very high proportion of wild harvesting populations (91.2%), a moderately higher proportion of livestock area (54.4%) and a much lower proportion of farmland (8.0%; of which 87% are small farms). We complement this profiling in Supplementary Information with assessments for land-use strategy of the overlapped population, with the ITT scenario highlighted as overlapping more people with smallholder strategies (Supplementary Fig. 8).

We also examined the development status and livelihoods of populations in areas where scenarios overlap. These overlap areas generally show intermediate conditions between the scenarios they connect (Supplementary Fig. 4).

## Spatial distribution between continents and scenarios

We assessed the broad distribution of population and HDI status among scenarios at the continental level (Fig. 3). Data quality was not sufficient to support country level analysis. The Biodiversity and NCP scenarios would more than double resident populations within protected and conserved areas compared to current levels on every continent. The Biodiversity scenario overlaps the largest population across all continents, with Asia having the highest total at 1030 million people—six times the population currently residing in protected and conserved areas there. This scenario also overlaps substantially larger populations in other regions, such as an 11-fold rise in North and Central America and a 27-fold rise in Australia and Oceania. In contrast, the ITT scenario adds fewer people than both other scenarios and current protected and conserved areas across all continents. Across most continents, populations within Target 3 scenario areas tend to have mean HDI values lower than the continental average, although this pattern is not consistent across all scenarios or regions (Fig. 3 and Supplementary Table 4). There is considerable variability between scenarios, with the ITT and NCP scenarios more frequently overlapping with populations with lower

mean HDI than the continental mean, whereas the Biodiversity scenario more frequently overlaps with populations with higher mean HDI, particularly in South America, Africa, Australia and Oceania. The uneven distribution between continents of the population and development status of people resident in areas included in our scenarios raises the possibility of inequitable distribution of costs and benefits of area-based conservation around the world[40]. It is highly likely that similar disparities would exist within countries were suitable datasets available to conduct that analysis.

## Discussion

Our results demonstrate that Target 3 is an immensely ambitious social as well as ecological target, highlighting that social factors at all scales will be critical for successful implementation. Our scenario analysis shows how different approaches to expanding protected and conserved areas vary in the number, levels of development and livelihoods of people living in additional areas, shaping the social challenges that Target 3 implementation may face.

Each of our scenarios raises particular social issues. Implementation along the lines of the Biodiversity and NCP scenarios would affect 100s of millions of people living in or near additional areas, many of whom are involved in farming, raising questions about potential trade-offs with food production[55,56]. The very large local population in the Biodiversity scenario could result in high costs of implementation and increased levels of human-wildlife conflict, although it might also create new opportunities for wildlife tourism and other local benefits. The NCP scenario includes large areas of relatively intact tropical forest that are needed to support ecosystem processes on which all people depend, but have small local populations, raising questions about the need for international support from countries with high demand for these services and the resources to pay for them. Far fewer people live in or near additional areas in the ITT scenario, but a high proportion of those that do are of low development status and highly dependent on biodiversity for their livelihoods. Understanding of the social conditions in our scenario areas should inform where and how to implement Target 3. However, in all cases, governance and management arrangements should be tailored to the local circumstances, remain consistent with the quality elements of the target, and include careful consideration of any areas that may not be suitable for inclusion[19,23]. Here we emphasise four key implications of our results.

First, while our scenarios serve to illustrate the range of social contexts in which Target 3 could be implemented, they do not reveal a socially "best" or "optimal" scenario. This is because choices about what to prioritise depend on value positions, of which there are many relating to area-based conservation[32,57]. Target 3 also needs to be considered alongside other spatially relevant targets with which there may be interactions or trade-offs (such as KMGBF Targets 1, 2, 4, 10 and 11), as well as the cross-cutting equity and rights targets (22 and 23), the Sustainable Development Goals and wider societal objectives.

Second, the actual impacts of expanded protected and conserved areas on people (and biodiversity) will depend on the specific management strategies and governance arrangements employed at each site[23,37]. For example, the NCP scenario would only deliver intended benefits to local people if it enables them to access flows of nature's contributions that are important to them[44]. The ITT scenario must be based on locally-led and inclusive governance, given that it comprises areas that are already Indigenous and traditional territories[31]. We have not attempted to model scenarios for management and governance or their social outcomes due to limitations with available data[58] and because Target 3 implementation in practice includes multiple and coexisting options that do not follow a single scenario. However, our results point to an inherent tension in the ability to achieve all the elements of Target 3. The number and socioeconomic status of people living within each of our scenario areas suggest that for Target 3 to

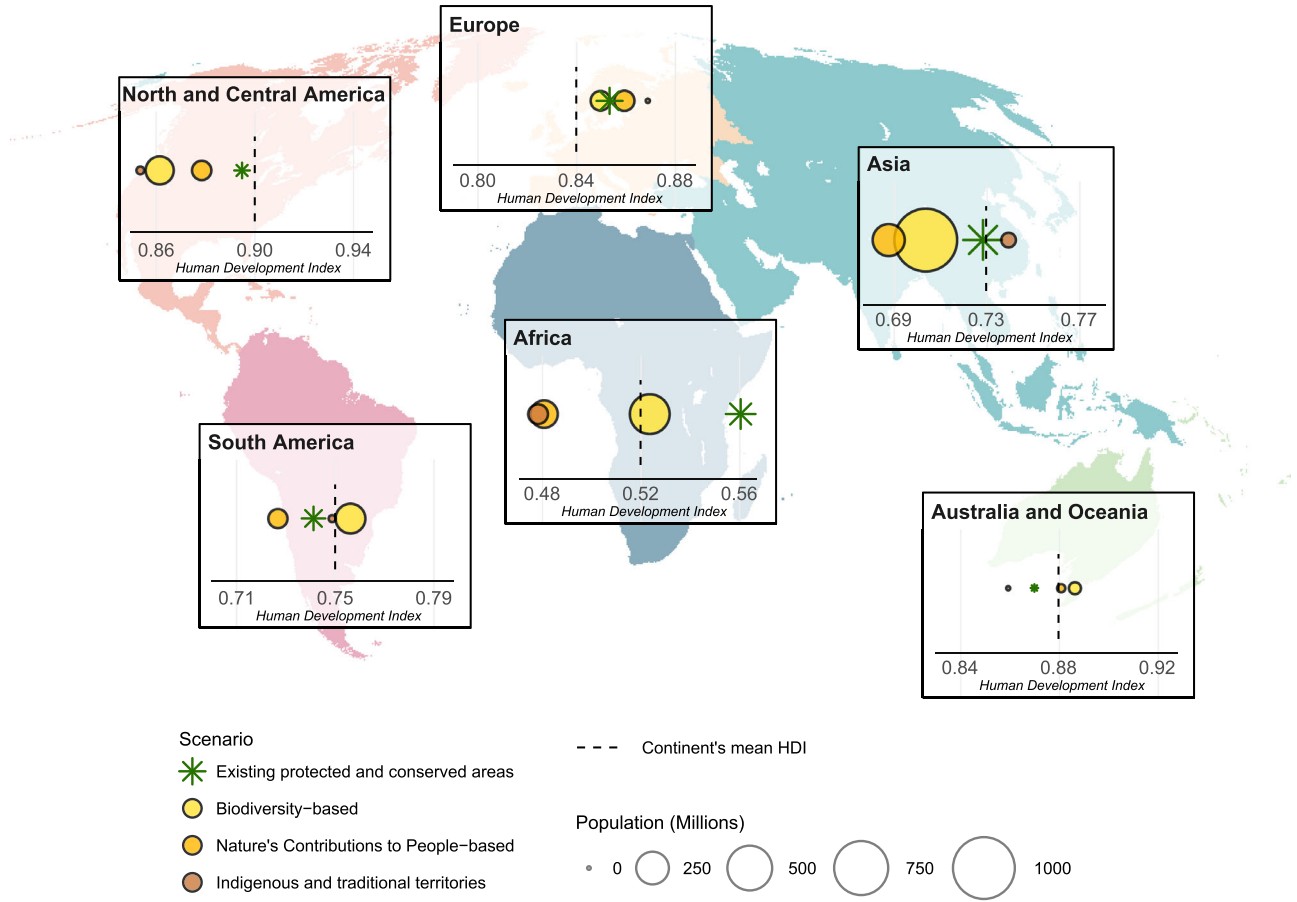

**Fig. 3 | Resident population and average Human Development Index (HDI) of populations in existing protected and conserved areas compared to new areas under Target 3 scenarios, by continent.** The size of the green asterisk represents the total resident population within existing protected and conserved areas, while the size of the circles indicates the resident population in areas included under each Target 3 scenario. The position of symbols along the x-axis reflects the average HDI of the population residing in the respective areas. HDI values are centred around the mean HDI of the total population in each continent, represented by a vertical dashed line.

achieve its social elements, most areas covered by newly established or recognised terrestrial protected and conserved areas will need to avoid strict, top-down forms of protection (such as forced relocation) that do not adequately integrate social considerations. However, to achieve some or all of the biodiversity elements required by the Target, strict protection in some cases may indeed be necessary[59–61]. Ultimately, site-level implementation needs to be informed by the local social and ecological context, and evidence regarding the likely outcomes for people and biodiversity of different management and governance options[22,23,37].

Third, our results demonstrate the value of incorporating social datasets into conservation mapping and prioritisation analyses, an approach that has not yet been widely adopted. We encourage others fully to integrate social datasets into conservation planning, not only as constraints but also as informative inputs into prioritisation. Such datasets can, for instance, be used to: (i) capture and contextualise the spatial patterns of human needs and capacities, thereby linking conservation action to human well-being[62]; (ii) reflect the complexity of land-use systems and actors by moving beyond coarse land-cover proxies to account for diverse practices and pressures, such as small-holder farming mosaics or forest-based livelihood activities[63]; or (iii) identify governance conditions and enabling environments that shape where conservation is socially feasible and likely to be effective[64]. This integration should include the local and national levels, where other relevant data, including Indigenous and local knowledge, may be available[65]. At the same time, we recognise the limitations of currently available data, the risks of over-interpretation, and the risk of data misuse, particularly associated with maps of ITTs[7,66–68].

Tracking the social impacts of Target 3 implementation also requires relevant social variables to be incorporated into the monitoring framework for the KMGBF, which currently only includes two social indicators out of 15[69]. Recognising that the development of measurable and verifiable social indicators remains challenging, particularly regarding data availability and consistency, the framework could nonetheless be strengthened by drawing on existing indicators already approved under cross-cutting Targets 22 and 23. These include, for example, indicators on tenure rights in Indigenous and local community territories and on participation, access to information and access to justice, which could be adapted to the context of protected and conserved areas[70]. Completion of site-level social impact assessments of protected and conserved areas[71] could also complement these efforts to track progress towards equitable governance. Lastly, further attention to and development of metrics for broader socio-economic outcomes, such as well-being and impacts on livelihoods[72], should be a priority within the rights-based approach reflected in the KMGBF preamble. Even if these are not suited for inclusion within the monitoring framework as indicators of Target 3 elements, they will be essential to ensure that future area-based conservation delivers positive outcomes for people as well as nature.

Finally, implementation of Target 3 requires a social development programme that matches the ambition of the target. A particular priority should be support for local residents to govern and manage

biodiversity at the local level and to minimise trade-offs with local livelihoods. This programme needs to include substantial funding, which could be supported through the GEF Global Biodiversity Framework Fund and other sources, and the involvement of civil society. Given that actions for the achievement of Target 3 will be implemented at the national and subnational level, it is essential that mechanisms be developed to provide funding and other support to countries and regions which will bear a higher proportion of the costs (monetary and non-monetary) of implementation. Activities might include compensation or guaranteed basic income[73] for those who are successfully conserving biodiversity and/or being negatively impacted by protected and conserved areas, funding for locally led conservation, and support for rights-based approaches and long-term monitoring and evaluation[18]. These need time and resources for processes like free, prior and informed consent to be carried out, meaning that a rush to achieve the spatial element of the target by 2030 at the expense of the social elements should be avoided.

## Methods

To evaluate the potential social implications of implementing Target 3 in terrestrial and inland water areas, we assessed social conditions at sites likely to be selected under three global scenarios. Below, we describe the scenarios, the spatial analysis approach to identify and profile populations within these areas, and the socio-economic indicators used to assess local social conditions.

### Current protected and conserved areas

We obtained global data on terrestrial PAs and OECMs from the August 2024 version of the World Database on Protected Areas (WDPA) and the World Database on Other Effective Area-based Conservation Measures[9]. We followed Protected Planet's three-step guidelines for filtering areas that count towards Target 3 reporting[74]. First, we excluded PAs with "Proposed" status and UNESCO-MAB Biosphere Reserves. MAB core areas typically overlap with national-level PAs, but their buffer and transition zones, often included in WDPA, are often unprotected, justifying the exclusion. All OECMs in the database were included. Second, to account for PAs with known areas but lacking spatial polygons, we generated circular buffers around centroids based on reported area. Point-format PAs without reported areas were discarded. Third, since our focus is on terrestrial and inland water regions, we removed marine PAs and OECMs, as well as marine portions of coastal areas.

Additional data sources were used for China and India, where data-sharing restrictions limit public WDPA access. For China, we incorporated PA data from Shen et al.[42], accessed in raster format, matching the resolution and projection of our dataset. For India, we used September 2019 Protected Planet data, the latest available complete dataset. Indian PAs were processed similarly to the global database. Datasets for China and India were then merged into the global dataset.

The global dataset in vector format was rasterised at 25 km² resolution (Mollweide projection) by calculating the proportion of each cell under PAs and OECMs. Cells with ≥50% coverage were classified as protected/conserved, a threshold that best matched the reported global coverage of PAs and OECMs[9]. Data were processed in R[75] using the "terra" package[76]. The combined dataset, referred to as current protected and conserved areas, covered 17.2% of Earth's terrestrial surface−0.3% lower than the global estimate from Protected Planet for the same period−reflecting residual data gaps for some PAs in China and India that were unavailable in the public WDPA and supplementary sources we consulted.

### Target 3 scenarios

**We examined three Target 3 scenarios.** (i) a *Biodiversity-based* approach that maximises biodiversity representation, (ii) a *NCP-based*

approach that maximises representation of areas that contribute local and global services to people, and (iii) an *Indigenous and traditional territories (ITT)* approach that prioritises ancestral lands governed by Indigenous Peoples and custodian communities with high biodiversity value. Each scenario included 12.8% of additional terrestrial and inland water areas, combined with the 17.2% of existing protected and conserved areas to achieve 30% coverage. While not predictive of actual implementation, these scenarios provide a framework to explore Target 3's social implications. Scenarios were refined to align with a common grid of consistent resolution (25 km²) and projection (World Mollweide), and the updated PA and OECM dataset. Analyses were conducted in R[75] using the "terra" package[76].

All maps were generated by the authors in R. Country boundaries shown in Fig. 1 and Supplementary Figs. 2 and 5 are for geographic reference only and were not used as analytical units. Boundaries were derived from United Nations–compliant boundary datasets, with disputed and special-status boundaries symbolised following UN cartographic conventions. Continental boundaries used in Fig. 3 follow the UN geoscheme.

**Biodiversity-based scenario.** This scenario prioritises the representation of species, ecoregions, and Key Biodiversity Areas (KBAs). It adapts results from Shen et al.[42], who used "prioritizr"[77] and a minimum shortfall objective function[78] to optimise unprotected area selection. Their optimisation included all unprotected KBAs[79], ≥17% representation of ecoregions[80], and maximising representation of 30,635 terrestrial vertebrates[81] based on "area of habitat" (AOH) maps[82], with species-specific representation targets informed by AOH extent[83].

We reproduced Shen et al.'s[42] results using data and scripts, updating PAs and OECMs to the current dataset. To reflect post-Aichi target priorities, we adjusted ecoregion representation targets from 17 to 15%, informed by sensitivity analyses of scenario layer selection in the biodiversity group (Supplementary Fig. 8), while retaining species-specific targets. Spatial prioritisation was conducted using 25 km² planning units, with PAs, OECMs, and KBAs locked in, the Gurobi solver[84], and a 1% optimality gap.

**Nature's contributions to people-based scenario.** This scenario identifies areas critical for delivering NCPs to the global population. Based on Neugarten et al.[44], it includes areas providing ten key NCPs: nine local-to-regional benefits (e.g., coastal risk reduction, water quality regulation, crop pollination, timber production) and one global benefit (vulnerable ecosystem carbon storage). The prioritisation used realised benefits, accounting for direct provision (e.g., fodder production per unit area) or weighting by beneficiaries (e.g., flood regulation by population downstream). Spatial optimisation used the "prioritizr" package with a minimum-set objective function[85]. Complementary to existing protected and conserved areas, their methods rank priority areas for NCP protection from highest to lowest NCP value.

We accessed Neugarten et al.'s priority ranking and updated protected and conserved areas data to the current version, assigning the highest priority value to these areas. The 2 km² resolution NCP data were resampled to 25 km² resolution by averaging pixel values, ensuring consistency with other scenarios. From this, the top 30% of highest priority terrestrial and inland water areas were selected, accounting for 70% of total NCP values.

**Indigenous and traditional territories scenario.** This scenario prioritises territories and areas governed by Indigenous peoples and custodian communities, which are estimated to support biodiversity conservation, often known as Indigenous and Community Conserved Areas (ICCAs) or territories of life. We based this scenario on UNEP-WCMC and ICCA Consortium's methodology for mapping potential ICCAs[45], updated with recent data.

First, we accessed spatial data for 270 ICCAs from the ICCA Registry[86]. For point-coordinate ICCAs, we generated buffers around centroids proportional to the reported area. Second, ICCAs were integrated with Indigenous Peoples and local communities' lands data from WWF et al.[14]. Although this dataset is the most comprehensive available, it represents an underrepresentation of the true extent of these lands. We replaced Australia's dataset with higher-resolution data from Jacobsen et al.[87]. Non-Indigenous areas and those "subject to other special rights" were excluded. Third, we combined these datasets with updated PAs and OECMs. Fourth, all data were rasterised to a 25 km² resolution.

Fifth, to prioritise areas with high biodiversity value, we excluded pixels with a Human Modification Index (HMI) > 0.1[88]. This threshold, also used by UNEP-WCMC and ICCA Consortium[45], reflects minimal anthropogenic pressures compatible with Indigenous and traditional land uses. Although the HMI is a pressure-based rather than ecological condition metric, it provides a globally consistent proxy for identifying areas with low human disturbance. Using it in combination with the layer representing lands of Indigenous peoples and local communities allows us to restrict the scenario to the parts of these lands that meet this low-modification criterion, while not treating the forms of land use by these groups that can sustain or enhance biodiversity as degradation[45].

Following these steps, the resulting layer covered 36% of global terrestrial and inland water areas. To align with the scenario's 30% target, pixels were randomly removed, except from ICCAs or existing protected and conserved areas, until coverage was reduced to 30%. To account for variability introduced by random removal, we performed 100 replicates, averaging results across the maps. Supplementary Fig. 5 illustrates areas included across all replicates. As this scenario incorporates sensitive data on ICCA and IPLC boundaries, high-resolution reproduction of this data is avoided to ensure confidentiality. Instead, we present results as 1° × 1° degree resolution maps (Fig. 1 and Supplementary Fig. 5).

Our three scenarios align with a global implementation of Target 3, where the contribution of countries and regions varies. This means that the percentage of countries' areas in each scenario can exceed or fall short of 30%. Scenarios do not consider downgrading, downsizing, and degazettement of PAs (PADDD) and OECMs by 2030[89]. PADDD events reduce the extent and quality of PA networks, and therefore would demand for more additional conservation areas beyond the 12.8% in our scenarios to counter area losses. However, we did not attempt to model them due to their unpredictability.

To estimate populations residing near but outside scenario areas, we created a 10 km buffer around each Target 3 scenario. We also estimated the population within 10 km of current protected and conserved areas for comparison. 10 km distance was chosen as it represents the range within which most socioeconomic impacts are typically felt[22,23]. We also performed a sensitivity analysis for varying buffer distances (Supplementary Fig. 10). A description of datasets underlying each scenario is provided in Supplementary Table 5.

## Assessment of local social conditions

We conducted spatial analyses using zonal statistics to evaluate the relationship between Target 3 scenarios and datasets quantifying population, HDI, and livelihood characteristics of potentially affected communities, comparing results with those for current protected areas. These analyses were performed using global raster datasets described below. Continental-level analyses were conducted by cropping these datasets using continental boundaries defined according to the United Nations geoscheme and applying the same analytical methods as for global-level results. Country boundaries were not used as analytical units in either the global or continental analyses.

All analyses used raster data aligned to a common 25 km² resolution grid in the Mollweide equal-area projection, ensuring consistency with the scenarios and protected and conserved area datasets. Spatial analyses were conducted in R[75] using the "terra" package[76].

**Human population analysis.** We quantified the population within and around Target 3 scenarios using the Global Human Settlement Population raster (GHS-POP)[46], which harmonises global census data (GPWv4.11) and built-up area distribution as mapped by GHSL[90]. The original raster data at 1 km² resolution were aggregated to the common 25 km² grid by summing the population counts within each cell. Population counts were calculated for 2023 and 2030 datasets. The 2030 dataset is a projection that assumes no population changes within the boundaries of existing PAs at the time of creation of the dataset, limiting its reliability for these regions. Supplementary Tables 1 and 2 report numerical results of the human population analysis across all scenarios, globally and by continent.

**Human development analysis.** We assessed the development status of affected populations using the MOSAIKS gridded HDI[47] (Version 2), a composite metric reflecting income, education, and health[91]. Global municipality-level HDI data for 2020 were rasterised at 25 km² resolution. Missing data for Libya were supplemented using 0.1° gridded estimates from the same source. HDI was categorised into four levels—low (< 0.55), medium (0.55–0.70), high (0.70–0.80), and very high (> 0.80)—following UNDP thresholds. We estimated the population in each HDI category within scenarios by overlaying the categorised HDI layer with gridded population data for 2023 (GHS-POP), assigning population counts to HDI categories based on the HDI classification of each pixel. The resulting layer was then aggregated to the common 25 km² grid by summing the population counts within each HDI category. Supplementary Table 2 reports the full numerical results of the human development analysis across all scenarios.

**Livelihood analysis.** We examined livelihoods potentially impacted by Target 3 scenarios using three datasets:

(1) Wild harvesting population: We estimated the population from wild harvesting households potentially affected using high-resolution maps of tropical wild harvesting[48]. This layer was generated using a spatially explicit dataset of more than 10,000 households representative of diverse peri-urban and rural areas across the tropics in 2015, and generalised with Bayesian modelling. Data were aggregated from the original 1 km² resolution to the common 25 km² grid by summing population counts and analysed for overlap with Target 3 scenarios to estimate affected populations. It is important to note that the wild harvesting data from Wells et al.[48] cover only tropical areas between 24°N and 24°S latitude.

(2) Farm area: Overlap with farmed areas was assessed using global maps of average farm size[49], focusing on small farms. This data uses crowd-sourced field size data[92] to classify global cropland area into farm size categories. The dataset was aggregated from 1 km² to the common 25 km² grid by summing the area (km²) within each farm-size class, and analysed for overlap with the classes not-farmed, small farms (< 5 ha), and larger farms.

(3) Livestock rangeland area: We calculated the overlap of Target 3 scenarios with livestock keeping production system areas[93]. Using data from the Global Livestock Production Systems (version 5)[50], we subsetted "livestock only systems" areas where livestock is more likely to be raised for subsistence or local sales rather than to supply larger markets. Specifically, we excluded mixed systems, both rainfed and irrigated[93], and resampled the data from its original ~100 km² resolution to the common 25 km² grid, retaining the modal production system. Supplementary Table 4 reports the full numerical results of the livelihoods analysis across all scenarios.

## Sensitivity analyses

To address uncertainties in socio-economic datasets and methods, we conducted five sensitivity analyses to evaluate the robustness of our findings under different assumptions and data sources. Results are provided in the "sensitivity analysis" section of Supplementary Information.

**Population datasets.** Modelled global gridded population products derived from census data may fail to accurately estimate population counts at different scales[94], especially in rural regions with low infrastructure and light, often targeted for conservation. To address these uncertainties, we repeated analyses using:

(1) WorldPop (2020): Constrained and unconstrained versions of the dataset at 100 m² and 1 km² resolution, respectively[95].

(2) LandScan[96] (2020).

(3) Gridded Population of the World[97] (GPW).

These datasets differ in methodology and assumptions, but are all top-down modeling performed at the global scale, and thus they may suffer from common uncertainties. The construction of WorldPop-related datasets involves data on existing PA boundaries to train and project models. This may raise circularity issues to assess the population within existing PAs, but should not affect estimations for additional Target 3 scenario areas.

**Local wealth conditions.** While HDI is widely used, there are other metrics for local wealth, each with distinct strengths and weaknesses. MOSAIKS' municipality-level gridded HDI[47] provides finer-scale data than previously available, but has not yet been peer-reviewed, which makes this sensitivity analysis more necessary. We compare our results with three additional commonly used metrics:

(1) Relative Wealth Index[98] (RWI). This gridded RWI results from machine-learning modelling trained on household survey data to estimate subnational wealth based on diverse inputs, including satellite imagery, mobile phone networks, topographic maps, and anonymized connectivity data. RWI emphasizes wealth-related aspects, which may not fully capture multidimensional well-being. Available for 135 low- and middle-income countries.

(2) International Wealth Index[99] (IWI). IWI is an assets-based composite index, which predicts village-level poverty using machine-learning models trained on demographic and health survey data on household asset ownership, using geospatial data such as satellite imagery, nighttime lights, and OpenStreetMap data to project wealth values over sub-Saharan African countries. IWI emphasizes asset-related aspects of wealth and may not fully capture multidimensional well-being.

(3) Human well-being derived from unlit settlement footprints[100]. This metric is derived from combining satellite nighttime lights and settlement footprints, enabling the estimation of well-being in areas lacking detectable artificial radiance—regions often missed by other nighttime radiance-derived metrics. This provides valuable insights into rural areas that could be designated as protected or conserved. Available for 49 countries across Africa, Asia and the Americas.

The three alternative metrics have limitations[66], including being only available sub-globally, but they are comparable in scope to HDI and provide valuable comparisons to assess the consistency of our HDI findings. For each metric, we obtained raster data for the full geographic extent where data were available (countries and territories are listed in Supplementary Table 6) and aligned them to the common grid and projection used in our analyses. To ensure comparability, HDI-based results were recalculated for the same country subsets corresponding to each dataset. As in the global analyses, zonal statistics were computed over the full area available rather than using country boundaries as analytical units.

**Livelihoods.** We compared our livelihood analysis with the global land-use strategy model by Malek and Verburg[101]. This dataset estimates the most likely livelihood-related land-use strategies at a given location by modelling biophysical, soil, and socio-economic contextual conditions, integrating data from a literature review and global datasets, including five biophysical variables, soil properties informing agronomic efficiency, and ten socio-economic indicators. Among the available strategy classes, we focused on survivalists, subsistence-oriented smallholders, and market-oriented smallholders, as the groups whose livelihoods are intricately tied to land use and access, corresponding with those who may be particularly sensitive to changes. To estimate the population potentially impacted within each strategy, we assigned population estimates from GHS-POP 2023 to strategies in 1 km² pixels, later aggregating data to 25 km² matching scenarios.

**Target 3 scenario layer selection.** Projecting areas to be designated under Target 3 involves considerable uncertainty. Our scenarios highlight areas with high potential to meet KMGBF criteria, but spatial representations of these areas vary with use of different datasets and methodologies by different authors. To assess uncertainties regarding the dataset used under each scenario category, we explored the following variations:

(1) Biodiversity-based scenario. We compared results obtained with the main scenario layer with those resulting from modified Shen et al.'s[42] prioritisation and from an alternative priority area mapping for biodiversity representation. Modified Shen et al. prioritisations consisted in maps generated with varying ecoregion representation targets, ranging from 0% (no optimisation for ecoregions) to 25%. The alternative biodiversity prioritisation used priority areas mapped by Jung et al.[102]. This layer is based on prioritisation methods consistent with Shen et al.[42], but incorporates a broader species set (including plant data) and different representation targets. It does not consider ecoregions and KBAs. Jung et al.[102] ranked planning units by priority rather than explicitly identifying a 30% coverage. The highest-priority areas encompassing 30% of terrestrial and inland water areas were identified, updated to our dataset of current protected and conserved areas, resampled, and analysed consistently with our main scenario.

(2) NCP-based scenario. For this scenario, we conducted sensitivity analyses using two additional optimisations: (1) an optimisation from Neugarten et al.[44] using data at a higher 25 km² resolution, and (2) an optimisation by Chaplin-Kramer et al.[103], which prioritised areas at the national level based on 10 local-scale and two global-scale NCPs. Additionally, we compared our results with a further optimisation by Neugarten et al.[44] that incorporates representation of 26,709 terrestrial vertebrate species alongside NCPs. This optimisation lies conceptually between the Biodiversity-based and NCP-based scenarios, providing a relevant point of comparison for both that scrutinises the impact of combined optimisation objectives. The three alternative layers share underlying data with the main scenario, except for the vertebrate species distribution data and two additional NCPs in the Chaplin-Kramer et al. layer (riverine fish harvest and atmospheric moisture recycling). Chaplin-Kramer et al.'s prioritisation achieves 30% equitable NCP coverage at the country-level, departing from other scenarios providing global-scale operationalisation of the target. All layers were processed using the same steps as the primary scenario.

(3) ITT scenario. This analysis included 99 additional replicates of the random removal process used in the primary scenario to test its robustness. Additionally, we compared the results with another set of 100 alternative scenario layers derived from Indigenous territories data by Garnett et al.[104], instead of the IP and LC lands dataset from WWF et al.[14]. All alternative layers followed the same analysis methods, including updates for Australian IP and LC lands[87]. Finally, we compared results with those obtained when all IP and LC lands mapped by

WWF et al.[14] rather than a random selection of the area. This approach results in a layer with a 36% global coverage instead of 30%.

**Scenario neighbouring areas.** Empirical evidence suggests that most positive and negative impacts occur within 10 km of PAs[23]. To account for variability in impacts extending beyond this distance, we estimated populations within 5 and 15 km of scenario areas.

## Reporting summary

Further information on research design is available in the Nature Portfolio Reporting Summary linked to this article.

## Data availability

This study did not generate or collect any new data. All data used in the analyses were obtained from publicly available sources and are cited in the manuscript. Readers can access the original datasets through the referenced sources.

## Code availability

The R scripts used for data processing, analysis, and figure generation in this study are publicly available at https://doi.org/10.5281/zenodo.18344001

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

## Acknowledgements
This research was conducted with funding and support from the Science for Nature and People Partnership (SNAPP) project "The Social Implications of 30×30". We are thankful to all SNAPP working group members, including those not participating as coauthors, for their contributions to discussions informing this research. We also thank Jessica Ball for facilitating several SNAPP working group workshops, and Christian Levers, Ryan Unks and Patrick Meyfroidt for advice on social spatial data. Funding support was provided by SNAPP grant SNP056 (J.F., C.S.), the ESRC-funded "Just Earth Observation for Conservation" project grant ES/Y002660/1 (C.R., R.P.), the European Research Council under the European Union's Horizon 2020 research and innovation programme through the CONDJUST grant 101054259 (D.B., J.F.) and the SYSTEM-SHIFT grant 101001239 (T.K.), the OECD Co-operative Research Programme: Sustainable Agricultural and Food Systems (2023) (F.F.), the UKRI/NERC grant SECO, NE/T01279X/1 and an APEX award from the Leverhulme Trust and Royal Society, APX/R1/191094 (C.M.R.), the Jameel Observatory in Food Security Early Action (G.R.W.). Support was also provided by the Centre for Sustainable Forests and Landscapes at the University of Edinburgh (C.M.R., G.R.W.). This work contributes to the Global Land Programme (https://glp.earth) and to ICTA-UAB "María de Maeztu" Programme for Units of Excellence of the Spanish Ministry of Science, Innovation and Universities (CEX2024-001506-M funded by MICIU/AEI/10.13039/501100011033). Views and opinions expressed are however, those of the authors only and do not necessarily reflect those of the European Union or the European Research Council Executive Agency. The boundaries and names shown and the designations used on maps do not imply official endorsement or acceptance by the United Nations.

## Author contributions
J.F., H.C.B., D.B., R.C.-K., J.A.F., F.F., A.F., R.G., C.H., T.K., J.L., M.O.F.N., B.O., F.O., M.P., R. Pinto, R. Pritchard, P.S., J.C.T., J.U., C.M.R., D.M.T., G.R.W., J.G.Z. and C.S. conceived the paper and developed the methodology. J.F., G.R.W. and C.S. coordinated data acquisition. J.F., T.K., J.L., C.M.R., G.R.W. and C.S. led the data analysis and the interpreting of results. J.F. and C.S. led the writing and editing of the manuscript. H.C.B., D.B., R.C.-K., J.A.F., F.F., A.F., R.G., C.H., T.K., J.L., M.O.F.N., B.O., F.O., M.P., R. Pinto, R. Pritchard, P.S., J.C.T., J.U., C.M.R., D.M.T., G.R.W. and J.G.Z. provided conceptual and technical input throughout the study and contributed to writing the manuscript. C.S. and D.B. secured financial support for the research project.

## Competing interests
The authors declare no competing interests.

## Additional information

[1]Department of Geography and Conservation Research Institute, University of Cambridge, Cambridge, UK. [2]Institut de Ciència i Tecnologia Ambientals de la Universitat Autònoma de Barcelona (ICTA-UAB), Cerdanyola del Vallès, Spain. [3]UN Environment Programme World Conservation Monitoring Centre, Cambridge, UK. [4]Department of Private Law, Universitat Autònoma de Barcelona, Cerdanyola del Vallès, Spain. [5]ICREA, Barcelona, Spain. [6]WWF, San Francisco, CA, USA. [7]The Nature Conservancy, Carlton, VIC, Australia. [8]School of Life and Environmental Sciences, Deakin University, Burwood, VIC, Australia. [9]School of Law, University of Tasmania, Sandy Bay, TAS, Australia. [10]Department of Forest Resources, University of Minnesota, St Paul, MN, USA. [11]Rights and Resources Initiative, Washington, DC, USA. [12]The Nature Conservancy, London, UK. [13]Geography Department, Humboldt-University Berlin, Berlin, Germany. [14]Integrative Research Institute on Transformations in Human-Environment Systems (IRI THESys), Humboldt-University Berlin, Berlin, Germany. [15]Micaia Foundation, Chimoio, Mozambique. [16]Campaign for Nature, Durango, CO, USA. [17]Private Consultant, Kampala, Uganda. [18]International Institute for Environment and Development, London, UK. [19]SNES, Newcastle University, Newcastle Upon Tyne, UK. [20]Global Development Institute, University of Manchester, Manchester, UK. [21]University of Edinburgh, School of Geosciences, Edinburgh, UK. [22]The Nature Conservancy, Arlington, VA, USA. [23]Global Youth Biodiversity Network, Panama City, WY, USA. [24]Partners for Indigenous Knowledge Philippines, Baguio City, Philippines. [25]Department of Environmental Management, Makerere University, Kampala, Uganda. [26]Wyss Academy for Nature, Bern, Switzerland. [27]Centre for Development and Environment and Institute of Geography, University of Bern, Bern, Switzerland. ✉e-mail: fajardonjavier@gmail.com; cgs21@cam.ac.uk

