## [Transparent Peer Review file · Nature Communications]

Social implications of the 30x30 global conservation target

Corresponding Author: Dr Javier Fajardo

Version 0:

Reviewer comments:

Reviewer #1

(Remarks to the Author)

This is a highly original and interesting paper that evaluates the population size, human development status, and livelihoods of people residing in areas that may be included under three different scenarios, namely the Biodiversity scenario, Nature's Contributions to People scenario, and the Indigenous and Traditional Territories scenario, for achieving the 30% coverage goal of Target 3 under the Kunming-Montreal Global Biodiversity Framework. I consider this study both timely and meaningful, offering valuable social insights into protected area expansion aimed at meeting the 30% coverage target. The authors identify potential protected and conserved areas under the three distinct scenarios and integrate a range of social datasets, analyzed using appropriate methodologies. One of the most commendable aspects of the study is the use of extensive sensitivity analyses to test the robustness of results, which significantly enhances the credibility and value of the study's conclusions. Below are several specific comments that I hope will help improve the manuscript:

Main text

Line 255: The analysis of wild harvesting focuses on the tropics. I recommend that this geographic limitation be explicitly acknowledged in the main text.

Line 190: The overlap between the Biodiversity and NCP scenarios is stated as 24%, but in Figure S3 it is shown as 23%.

Figure S3(B): There appears to be an error. According to the text above Figure S2, the area of overlap among the three middle scenarios should be 0.6%.

Lines 220–222: Based on Supplementary Table 1 and the text above Figure S5, if my calculation is correct, the population growth in the ITT scenario area by 2030 should be approximately $(27.5/120) \approx 23\%$, rather than 30%.

Line 228: "v" should be corrected to "vs".

Lines 270–272: It is difficult to determine the number of people residing in areas overlapping the Biodiversity and NCP scenarios solely based on Figure 3. The bubble sizes in the figure are not intuitive for estimating population sizes. I recommend adding a supplementary table summarizing the population sizes under different scenarios, disaggregated by continent.

Lines 278–280: The conclusion that populations under the Biodiversity scenario generally have a lower HDI than the continental average does not appear to hold in all cases. For example, in South America, Africa, and Australia and Oceania, the populations have mean HDI values higher than the continental average. Hence, this conclusion may not be broadly generalizable.

Discussion

Lines 329–330: The rationale for avoiding strict and top-down forms of protection to fulfill the social elements of Target 3 is not clearly explained. From a macro perspective, encouraging top-down protection mechanisms that adequately integrate social considerations might also be a valid approach to balancing social and biodiversity outcomes. I suggest the authors further elaborate on this point.

Lines 336–344: Integrating social datasets into systematic conservation planning is a highly insightful and compelling proposal. However, what roles do these social datasets actually play in the prioritization process? In many existing

conservation planning frameworks, social data are included but often only as opportunity costs—used, for instance, to ‘exclude’ areas of high agricultural productivity. The manuscript would benefit from a more in-depth discussion on the potential functions of social datasets beyond serving as constraints or costs, particularly how they can contribute proactively to prioritization.

Methods

Lines 587–589: According to the WDPA, not all marine PAs are purely marine; some also include coastal land areas. If such areas were excluded, could this lead to underrepresentation of the social conditions of coastal communities?

Lines 577–595: Was there a size threshold applied to the protected areas included in the analysis? How did the authors ensure that the spatial resolution of the PA data aligns with that of the social datasets?

Lines 597–667: The authors have done an excellent job delineating potential protected and conserved areas under three distinct scenarios. However, I have a small concern regarding the ITT scenario. When identifying areas of high biodiversity value, why was the Human Modification Index used instead of more relevant metrics such as Key Biodiversity Areas (KBAs) or AOH of species? Is the Human Modification Index a sufficiently robust proxy for assessing biodiversity value?

Reviewer #2

(Remarks to the Author)

The manuscript illustrates socioeconomic traits of population potentially impacted by the expansion of protected and conserved areas as per Target 3 of the GBF, by comparing three alternative scenarios. The paper is clearly written, methods are explained in details which make the results reproducible, and results are adequately supported by sensitivity analysis, though a certain level of uncertainty is intrinsic in scenarios. Scenarios and its implications are analysed by using among the best available datasets at global scale. Since the purpose of the paper is to outline the 'socioeconomic profiles' of areas where further PCA expansion may occur and consequent trajectories, the results are extremely significant even though the absolute statistics may vary and even improve through more local analyses and data.

- However, emerging overall trajectories should be better discussed in the context of the whole Global Biodiversity Framework and its 23 targets. In particular in relation to interactions/trade-off between Target 3, and T 2, 4, 10 and 11.

- Further, I suggest to enrich the discussion around the social implications of the scenarios. Currently, there is only one sentence referring to possible trade-offs with food production, while a large part of the discussion reports valid considerations on management and governance that may be done without this scenarios analysis. As stated by authors, achieving the complementary objectives of T3 may require a different approach than in the past, and this has different implications in regards of different social profiles.

Just a few thoughts that came to my mind, which are not meant to influence yours: It seems like biodiversity valuable areas are highly populated, and so targeting biodiversity only would make the challenges of human-wildlife coexistence even higher than in current situation. On the contrary, protecting NCP may not be recognised as "convenient" for societies in certain areas if the population, and therefore NCP demand, is low. This goes beyond the consideration on actual access to the NCP flow, it is more about whether decision makers would see the value of investing in such unpopulated areas. The low population of indigenous territories may represent a high risk for the future maintenance of such territories and the actual acknowledgement and respect of indigenous people rights; such risk, eg. of people eviction, may eventually turn against biodiversity conservation objectives.

-Then, the discussion suggests to improve tracking of social impact of Target 3. Maybe the author could add a few considerations on how the social profiles and trajectories may actually be changed by T3 implementation under each scenarios, in terms of opportunities and benefits.

- Past studies mapped priority areas for conservation even based on a mixture of dimensions (EG biodiversity, carbon, water or biodiversity and Nature Contribution to People). While presenting 'extreme' scenarios may facilitate the emergence of criticalities, it may be interesting to evaluate social implication for a mixed scenario as well, even though overlap between scenario is low. what are the socioeconomic profiles for the overlapping areas between scenarios pairs and the 3 of them? it might be interesting to add this to supplementary material and in the discussion if any relevant outcome is found.

Reviewer #3

(Remarks to the Author)

This manuscript tackles an important and very timely question: what are the potential social implications of achieving Target 3 (“30x30”) of the Kunming-Montreal Global Biodiversity Framework? Using three scenarios, the authors combine global datasets to explore how many people could be affected by 30x30, and in what socio-economic contexts. This is a valuable and much-needed global assessment of social variables in conservation planning, and it makes a strong and relevant contribution to ongoing policy and implementation debates. It fills a surprising ongoing gap in the 30x30 conversation.

The study is methodologically robust, original, and well suited for Nature Communications. I see only minor revisions that would strengthen the clarity of the work without altering the core analysis.

Minor suggestions:

The inclusion of the ITT scenario is important, but while the paper notes confidentiality concerns and uses coarse resolution to protect data, it could also briefly discuss:

- The contested nature of “recognition” within the CBD framework.
- The risk of data misuse or co-optation, especially if such maps are misused as definitive boundaries.

The point about the KMGBF monitoring framework containing only two social indicators is well made. The authors might consider recommending specific, feasible metrics that could be integrated to track social impacts of 30×30 implementation.

Terminology could be tightened by ensuring consistent use of “Indigenous and Traditional Territories” versus “Indigenous Peoples’ and Local Communities’ lands” (or clarifying the intended distinctions).

Finally, Figures 2 and 3 are wonderfully rich. Consider including a note to reiterate that the “wild harvesting” data apply only to tropical regions.

This is an excellent piece of work. I look forward to seeing it published soon.

Best wishes,

Steph

Version 1:

Reviewer comments:

Reviewer #1

(Remarks to the Author)

Thank you to the authors and editors for the opportunity to re-review this manuscript. I sincerely appreciate the authors' thoughtful responses to my previous comments and their diligent revisions. The added discussion on incorporating social datasets into conservation mapping and prioritization analysis is particularly valuable.

All of my previous concerns have been fully addressed, and I am pleased to recommend acceptance of this manuscript for publication in Nature Communications. I commend the authors for this important contribution to the field.

Reviewer #2

(Remarks to the Author)

Dear authors,

thank you for taking into accounts my suggestions and comments, in particular as concern the Discussion and the additional analysis presented in the Supplementary Material. I feel this is a very important contribution to the global debate to Target 3 achievement and I hope will inspire more regional and local scenario analyses.

Point-by-point response to reviewers

Manuscript title: Social implications of the 30x30 global conservation target

Manuscript ID: NCOMMS-25-32970A

Thank you for the opportunity to revise our manuscript. We are grateful for the reviewers' thoughtful and constructive comments, which have helped us to improve the manuscript substantially.

Below, we provide a point-by-point response to all comments, with the reviewers' original remarks reproduced verbatim in the first column and our responses in the second.

Reviewer comment	Response
Reviewer 1	
Line 255: The analysis of wild harvesting focuses on the tropics. I recommend that this geographic limitation be explicitly acknowledged in the main text.	We have incorporated explicit acknowledgements of the pantropical scope of the wild harvesting dataset in the abstract, introduction, results section, and Figure 2 caption. Specifically (new text is underlined): - In the abstract (line 61), we added: "high participation in tropical wild harvesting (91%)".- In the introduction (line 163), we added: "wild harvesting¹⁰ (in tropical areas)".- In the results (line 255), we revised to: "has a lower proportion of people living in the tropics who use wild harvesting".- In the Figure 2 caption, we added: "... (C1) wild harvesting status of households (tropical population) ...".
Line 190*: The overlap between the Biodiversity and NCP scenarios is stated as 24%, but in Figure S3 it is shown as 23%.	We thank the reviewer for noting this inconsistency. The discrepancy arose because, although the scenarios were designed to represent the same total extent, their precise pixel count differed slightly, as each was sourced from a different study. Consequently,

(*line 198 in the revised manuscript)	while the absolute overlap area between scenarios was identical, the percentage it represented varied marginally depending on the total area of each scenario. To resolve this, we recalculated the overlap proportions in Figure S3A using a common nominal reference area equivalent to 12.8% of global terrestrial extent—the intended new area for each hypothetical scenario complementing current protected and conserved areas. The revised percentages now represent the proportion of this nominal area that is unique to, or shared among, scenarios. The resulting values differ from the original ones only in decimals. We have updated the text at Supplementary Information (section “2. Overlap among Target 3 scenarios”), and the figure caption to reflect this clarification. These now explain that overlaps were computed using the terra and eulerr packages in R and expressed as proportions of the additional area in each scenario or relative to global land area (Figs. S3A). (Supplementary Information, line 104): “We used the ‘terra’ R package to calculate the areas that were unique to each scenario and those that overlapped between them (Fig. S2). We then generated Venn diagrams with the ‘eulerr’ package in R to visualise these overlaps, expressed either as percentages of the new nominal area of each scenario (i.e., 12.8% of the global area; the standardised target area assigned to each scenario, excluding current protected and conserved areas; Fig. S3A) or relative to the global land area (Fig. S3B). As a result of expressing overlaps as percentages of the new nominal area in each scenario (Fig. S3A), the summed proportions for a given scenario may not add exactly to 100%. This small deviation reflects the minor differences in total scenario areas but ensures direct comparability among them.” We also made minor corrections to overlap percentages described in the section to reflect the percentages with the newly used method. Finally, we modified the figure caption: “Figure S3: Venn plots showing overlap among Target 3 scenarios. (A) Percentage overlap between scenarios, showing the proportion of the new nominal area of scenarios, excluding current protected and conserved areas, that overlaps with the others”.
Figure S3(B): There appears to be an	We thank the reviewer for noting this. The apparent

error. According to the text above Figure S2, the area of overlap among the three middle scenarios should be 0.6%.	discrepancy arose because Figure S3B originally showed the total overlap of the three scenarios including the existing protected and conserved areas (PAs and OECMs), whereas the 0.6% value reported in the text referred only to the additional area of overlap among the scenarios outside PAs and OECMs. To clarify this, we have updated Figure S3B so that the area unique to the scenarios and the area overlapping with current PAs and OECMs are now shown separately. The text and figure descriptions have been amended accordingly. (Supplementary Information, line 129) “Areas shared by all three scenarios constitute only a small fraction - 4.6% of the new area designated by the scenarios - covering just 0.6% of the global terrestrial and inland water area (in addition to 17.2% corresponding to existing protected and conserved areas).” Finally, Fig S3B caption now explains: “The green area at the centre corresponds to existing protected and conserved areas, which are common to all three scenarios, while the outer portion of the central area represents additional overlap among the three scenarios outside these areas.”
Line 228*: "v" should be corrected to "vs". (*line 239 in the revised manuscript)	Thank you for noticing this typo. We have corrected it.
Lines 270–272*: It is difficult to determine the number of people residing in areas overlapping the Biodiversity and NCP scenarios solely based on Figure 3. The bubble sizes in the figure are not intuitive for estimating population sizes. I recommend adding a supplementary table summarizing the population sizes under different scenarios, disaggregated by continent. (*line 287 in the revised manuscript)	We thank the reviewer for this helpful suggestion. We have now added a supplementary table summarising the population residing within areas of overlap under each scenario, disaggregated by continent (Supplementary Table 4).
Lines 278–280: The conclusion that populations under the Biodiversity scenario generally have a lower HDI than the continental average does not appear to hold in all cases. For example, in South America, Africa, and Australia and	Thank you for this observation. We agree that the relationship between HDI values and Target 3 scenarios varies across regions and scenarios. Our intention was not to imply that all scenarios consistently overlap with populations below the continental mean, but rather to highlight that this is a frequent pattern,

Oceania, the populations have mean HDI values higher than the continental average. Hence, this conclusion may not be broadly generalizable. (*line 295 in the revised manuscript)	particularly for the ITT- and NCP-based scenarios. To clarify this, we have revised the corresponding paragraph (line 295) to read: “Across most continents, populations within Target 3 scenario areas tend to have mean HDI values lower than the continental average, although this pattern is not consistent across all scenarios or regions (Figure 3, Supplementary Table 4). There is considerable variability between scenarios, with ITT and NCP scenarios more frequently overlapping with populations with lower mean HDI than the continental mean, whereas the Biodiversity scenario more frequently overlaps with populations with higher mean HDI, particularly in South America, Africa, and Australia and Oceania.”
Lines 329–330*: The rationale for avoiding strict and top-down forms of protection to fulfill the social elements of Target 3 is not clearly explained. From a macro perspective, encouraging top-down protection mechanisms that adequately integrate social considerations might also be a valid approach to balancing social and biodiversity outcomes. I suggest the authors further elaborate on this point. (*line 363 in the revised manuscript)	This is a good point. We have edited the text as follows, to emphasise that what need to be avoided are strict, top down actions that do not adequately integrate social considerations (and we thank the reviewer for this turn of phrase). We have included forced relocation as an example of the kind of action to be avoided (line 363): “The number and socioeconomic status of people living within each of our scenario areas suggest that for Target 3 to achieve its social elements, most areas covered by newly established or recognised terrestrial protected and conserved areas will need to avoid strict, top-down forms of protection (such as forced relocation) that do not adequately integrate social considerations.”
Lines 336–344*: Integrating social datasets into systematic conservation planning is a highly insightful and compelling proposal. However, what roles do these social datasets actually play in the prioritization process? In many existing conservation planning frameworks, social data are included but often only as opportunity costs—used, for instance, to ‘exclude’ areas of high agricultural productivity. The manuscript would benefit from a more in-depth discussion on the potential functions of social datasets beyond serving as constraints or costs, particularly how they can contribute proactively to prioritization.	We thank the reviewer for this excellent point. We have edited the text to highlight the need to go beyond simple mapping of opportunity costs, as follows (line 373): “Third, our results demonstrate the value of incorporating social datasets into conservation mapping and prioritisation analyses, something which has been rare to date. We encourage others fully to integrate social datasets into conservation planning, not only as constraints but also as informative inputs into prioritisation. Such datasets can, for instance, be used to: (i) capture and contextualise the spatial patterns of human needs and capacities, thereby linking conservation action to human well-being⁶²; (ii) reflect the complexity of land-use systems and actors by moving beyond coarse land-cover proxies to account for diverse practices and pressures, such as

(*line 373 in the revised manuscript)	smallholder farming mosaics or forest-based livelihood activities⁶³; or (iii) identify governance conditions and enabling environments that shape where conservation is socially feasible and likely to be effective⁶⁴. This integration should include the local and national levels, where other relevant data, including Indigenous and local knowledge, may be available⁶⁵.”
Lines 587–589*: According to the WDPA, not all marine PAs are purely marine; some also include coastal land areas. If such areas were excluded, could this lead to underrepresentation of the social conditions of coastal. (*line 670 in the revised manuscript)	Our protected and conserved areas dataset includes the terrestrial portions of all PAs and OECMs, including these described coastal areas. In the WDPA, areas that overlap land and sea are labelled as ‘coastal’ (MARINE field = 1). For these cases, we clipped the polygons to retain only their terrestrial portions and excluded the marine parts from our analyses. We have revised the methods section (line 670) to clarify this point: “we removed marine PAs and OECMs, as well as marine portions of coastal areas”.

Lines 577–595*: Was there a size threshold applied to the protected areas included in the analysis? How did the authors ensure that the spatial resolution of the PA data aligns with that of the social datasets?

(*line 678 in the revised manuscript)

We thank the reviewer for this helpful comment. We have made corrections to include more comprehensive detail about these aspects.

We did not exclude any protected or conserved areas based on their size. Instead, we harmonised spatial data to a common 25 km² resolution raster template to ensure comparability across all layers. This resolution was chosen because it matches the coarsest of the scenario datasets (the biodiversity-based scenario) while remaining suitable for a global analysis of this type. The biodiversity-based scenario was generated from computationally intensive methods using input layers of varying resolution, making finer resolutions both impractical and unnecessary for our purposes.

Protected areas and OECMs, originally in vector format, were rasterised to this template by computing the percentage of each 25 km² grid cell covered by PAs/OECMs. To identify cells considered “protected/conserved”, we applied a 50% coverage threshold. This threshold was the one closest preserving the global extent of the PA and OECM networks as reported by the WDPA/WD-OECM while ensuring consistent alignment with other datasets. Importantly, this method does not eliminate small PAs or OECMs per se, as multiple small areas within the same cell can collectively exceed the threshold. The procedure also avoids inflating the global coverage due to marginal overlaps with PAs/OECMs that occupy only a small fraction of a cell.

The revised manuscript clarifies this as follows (line 678): “The global dataset in vector format was rasterised at 25 km² resolution (Mollweide projection) by calculating the proportion of each cell under PAs and OECMs. Cells with ≥50 % coverage were classified as protected/conserved, a threshold that best matched the reported global coverage of PAs and OECMs¹⁷.”

All other datasets (e.g., social indicators) were aggregated and resampled to the same 25 km² resolution to ensure spatial alignment. We made small adjustments to the relevant sections to make this clearer.

In line 698 (on scenarios): “Scenarios were adapted to align with a common grid of consistent resolution and projection, and the updated PA and OECM dataset”.

In line 787 (assessment of local social conditions): “All analyses used raster data aligned to a common 25 km² resolution grid in the Mollweide equal-area projection, ensuring consistency with the scenarios and protected

and conserved area datasets”.

Then, each specific analysis section now explicitly describes the data aggregation approach: line 795 for population count, line 812 for human development, and lines 823, 829, and 838 for the three livelihood datasets.

Lines 597–667*: The authors have done an excellent job delineating potential protected and conserved areas under three distinct scenarios. However, I have a small concern regarding the ITT scenario. When identifying areas of high biodiversity value, why was the Human Modification Index used instead of more relevant metrics such as Key Biodiversity Areas (KBAs) or AOH of species? Is the Human Modification Index a sufficiently robust proxy for assessing biodiversity value? (*line 747 in the revised manuscript)	We thank the reviewer for this thoughtful comment. The ITT scenario follows the approach proposed by UNEP-WCMC and the ICCA Consortium (2021), which identified “potential Territories of Life” by intersecting Indigenous peoples’ and local communities’ lands with areas of low Human Modification Index (HMI) values. We adopted this method to maintain consistency with the established ICCA mapping framework, updating it only with the most recent protected and conserved area layers (2024). This is in line with our overall approach of basing each scenario on previously published methods rather than developing new ones. We agree that the HMI is a pressure-based rather than an ecological condition metric. However, globally consistent alternatives for ecological condition remain limited, which supports the rationale for using the HMI in UNEP-WCMC and ICCA Consortium (2021). Its use in that publication reflects its ability to provide a conservative, globally consistent filter for identifying areas with low levels of anthropogenic pressures. In revising the manuscript, we have clarified this rationale and improved the explanation of how the HMI threshold relates to Indigenous and traditional land uses. The updated text in the Methods (line 747): "Fifth, to prioritise areas with high biodiversity value, we excluded pixels with a Human Modification Index (HMI) >0.1⁸⁸. This threshold, also used by ⁴⁹, reflects minimal anthropogenic pressures compatible with Indigenous and traditional land uses. Although the HMI is a pressure-based rather than ecological condition metric, it provides a globally consistent proxy for identifying areas with low human disturbance. Using it in combination with the layer representing lands of Indigenous peoples and local communities allows us to restrict the scenario to the parts of these lands that meet this low-modification criterion, while not treating the forms of land use by these groups that can sustain or enhance biodiversity as degradation⁴⁹".
Reviewer 2	
Since the purpose of the paper is to outline the 'socioeconomic profiles' of areas where further PCA expansion may occur and consequent trajectories, the results are extremely significant even though the absolute statistics may vary and even improve through more local analyses and data. However, emerging overall trajectories should be better discussed in	We have added this excellent point to the paragraph describing our first key take home message in the discussion, about the need to recognise that none of our scenarios emerges as ‘optimal’. This seems an appropriate point to highlight the wide range of relevant targets and goals and the possibility of interactions and trade-offs. The text now reads as follows (line 345, new words underlined): “First, while our scenarios serve to illustrate the range

the context of the whole Global Biodiversity Framework and its 23 targets. In particular in relation to interactions/trade-off between Target 3, and T 2, 4, 10 and 11.	of social contexts in which Target 3 could be implemented, they do not reveal a socially 'best' or 'optimal' scenario. This is because choices about what to prioritise depend on value positions, of which there are many relating to area-based conservation^{37,57}. Target 3 also needs to be considered alongside other spatially relevant targets with which there may be interactions or trade-offs (such as Targets 1, 2, 4, 10 and 11), as well as the cross-cutting equity and rights targets (22 and 23), the Sustainable Development Goals and wider societal objectives."
Further, I suggest to enrich the discussion around the social implications of the scenarios. Currently, there is only one sentence referring to possible trade-offs with food production, while a large part of the discussion reports valid considerations on management and governance that may be done without this scenarios analysis. As stated by authors, achieving the complementary objectives of T3 may require a different approach than in the past, and this has different implications in regards of different social profiles. Just a few thoughts that came to my mind, which are not meant to influence yours: It seems like biodiversity valuable areas are highly populated, and so targeting biodiversity only would make the challenges of human-wildlife coexistence even higher than in current situation. On the contrary, protecting NCP may not be recognised as "convenient" for societies in certain areas if the population, and therefore NCP demand, is low. This goes beyond the consideration on actual access to the NCP flow, it is more about whether decision makers would see the value of investing in such unpopulated areas. The low population of indigenous territories may represent a high risk for the future maintenance of such territories and the actual acknowledgement and respect of indigenous people rights; such risk, eg. of people eviction, may eventually turn against biodiversity conservation objectives.	We thank the reviewer for this helpful observation and for the thoughtful reflections on the different social implications that may arise under each scenario. We agree that the discussion benefits from making these links more explicit. In the revised version, we have expanded the opening paragraph of the Discussion to describe how each scenario relates to distinct social conditions and implications — including differences in population size, dependence on natural resources, potential trade-offs, and social risks — as suggested by the reviewer. The revised text (lines 322–344) now reads in part (new text underlined): "Our results demonstrate that Target 3 is an immensely ambitious social as well as ecological target, highlighting that social factors at all scales will be critical for successful implementation. Our scenario analysis shows how different approaches to expanding protected and conserved areas vary in who lives there, their levels of development, and their livelihoods, shaping the social challenges that Target 3 may face. Each of our scenarios raises particular social issues. Implementation along the lines of the Biodiversity and NCP scenarios would affect 100s of millions of people living in or near additional areas, many of whom are involved in farming, raising questions about potential trade-offs with food production^{55,56}. The very large local population in the Biodiversity scenario could result in high costs of implementation and increased levels of human-wildlife conflict, although it might also create new opportunities for wildlife tourism and other local benefits. The NCP scenario includes large areas of relatively intact tropical forest that are needed to support ecosystem processes on which all people depend but have small local populations, raising questions about the need for international support from countries with high demand for these services and the resources to pay for them. Far fewer people live in or near additional areas in the ITT scenario, but a high proportion of those that do are of low development status and highly dependent on biodiversity for their

	livelihoods. Understanding of the social conditions in our scenario areas should inform where and how to implement Target 3. However, in all cases governance and management arrangements should be tailored to the local circumstances, remain consistent with the quality elements of the target, and include careful consideration of any areas that may not be suitable for inclusion^{4,5}. Here we emphasise four key implications of our results.”
Then, the discussion suggests to improve tracking of social impact of Target 3. Maybe the author could add a few considerations on how the social profiles and trajectories may actually be changed by T3 implementation under each scenarios, in terms of opportunities and benefits.	The reviewer is inviting us here to speculate about how the implementation of Target 3 under our three scenarios could impact upon the lives of affected people. While we agree that this is a very interesting and important question, it is something that we decided not to do as part of this project. This is because it depends so much on how area-based conservation is implemented, and not just where. Without adequate data to conduct a meaningful analysis of possible impacts, we prefer not to make this kind of speculation, particularly given the sensitivities around debates over the impacts of area-based conservation. This is a key point of the paragraph at lines 353-372 in the discussion.
Past studies mapped priority areas for conservation even based on a mixture of dimensions (EG biodiversity, carbon, water or biodiversity and Nature Contribution to People). While presenting 'extreme' scenarios may facilitate the emergence of criticalities, it may be interesting to evaluate social implication for a mixed scenario as well, even though overlap between scenario is low. What are the socioeconomic profiles for the overlapping areas between scenarios pairs and the 3 of them? it might be interesting to add this to supplementary material and in the discussion if any relevant outcome is found.	We thank the reviewer for this excellent comment, which helped us strengthen the manuscript by introducing new analyses and clarifying the treatment of additional scenarios. We have made changes in two complementary directions. First, we conducted new analyses examining the population of areas where the main scenarios overlap and their socioeconomic characteristics. These results, now presented in Figure S4, show that overlap areas generally represent intermediate conditions between the scenarios they connect. The full description of methods and results for this new analysis can be found in lines 104-113 and 156-185, respectively (Supplementary Information). We also included a couple of sentences in the main manuscript pointing to these results and Figure S4 (line 267): “We also examined the development status and livelihoods of populations in areas where scenarios overlap. These overlap areas generally show intermediate conditions between the scenarios they connect (Figure S4, Supplementary Information).” Second, we clarified and made more visible our analysis of additional scenarios in the Supplementary

	Material, including a “mixed scenario”. In this sensitivity analysis (Figure S8), we evaluated ten additional scenarios. While some represent variations of the main scenarios, others reflect substantially different approaches. Among these is the “NCPs + species” scenario from Neugarten et al. (2024), which represents an intermediate—or “mixed”— approach combining biodiversity (26,709 terrestrial vertebrate species) and NCP layers. To make this scenario more distinct and identifiable as a “mixed” scenario, we have repositioned it in Figure S8 between the Biodiversity- and NCP-based groups. We also assigned it a mid-tone colour in the “A. Total population” portion of the figure to reflect its intermediate character. We chose not to expand the main analysis beyond the three principal scenarios to maintain clarity and avoid introducing new hybrid prioritisation methods. Our framework was intentionally designed to rely on approaches and data already established in the literature, ensuring that the methods used are widely recognised and do not require additional validation within this study. These three scenarios are illustrative of the range of different approaches that could be adopted, so while it is unlikely that any one of them will be implemented in full, they are helpful for exploring the implications of different choices about implementation.
Reviewer 3	
The inclusion of the ITT scenario is important, but while the paper notes confidentiality concerns and uses coarse resolution to protect data, it could also briefly discuss:  • The contested nature of “recognition” within the CBD framework. 	Thank you for this comment. The original manuscript did mention this point, but didn’t explicitly link recognition to ongoing debates about ITT areas. We have modified the text at line 91-93 so it now reads as follows (new words underlined): “However, how ITTs could be recognised and incorporated into Target 3 remains contested and is likely to vary within and between countries^{12,16}”
 • The risk of data misuse or co-optation, especially if such maps are misused as definitive boundaries. 	We have added this to the list of limitations and issues with social data in the discussion, at line 385. “At the same time, we recognise the limitations of currently available data, the risks of over-interpretation, and the risk of data misuse, particularly associated with maps of ITTs”.
The point about the KMGBF monitoring framework containing only two social indicators is well made. The authors might	We thank the reviewer for this constructive suggestion. We agree that proposing specific metrics would strengthen our discussion on how social impacts could

consider recommending specific, feasible metrics that could be integrated to track social impacts of 30x30 implementation.	be monitored under Target 3. We have therefore expanded the final paragraph of this section to include concrete examples of indicators that could complement those currently included in the KMGBF monitoring framework. The new text also refers to the feasibility of these indicators and to further development required in some cases. It reads as follows (line 388): “Tracking the social impacts of Target 3 implementation also requires relevant social variables to be incorporated into the monitoring framework for the KMGBF, which currently only includes two social indicators out of 15⁶⁹. Recognising that the development of measurable and verifiable social indicators remains challenging, particularly regarding data availability and consistency, the framework could nonetheless be strengthened by drawing on existing indicators already approved under cross-cutting Targets 22 and 23. These include, for example, indicators on tenure rights in Indigenous and local community territories and on participation, access to information and access to justice, which could be adapted to the context of protected and conserved areas⁷⁰. Completion of site-level social impact assessments of protected and conserved areas⁷¹ could also complement these efforts to track progress towards equitable governance. Lastly, further attention to and development of metrics for broader socio-economic outcomes, such as well-being and impacts on livelihoods⁷², should be a priority within the rights-based approach reflected in the KMGBF preamble. Even if these are not suited for inclusion within the monitoring framework as indicators of Target 3 elements, they will be essential to ensure that future area-based conservation delivers positive outcomes for people as well as nature.”
Terminology could be tightened by ensuring consistent use of “Indigenous and Traditional Territories” versus “Indigenous Peoples’ and Local Communities’ lands” (or clarifying the intended distinctions).	Thanks for highlighting this. We have checked the text carefully and made a few edits for consistency and clarity. A challenge here is that several of the secondary data sources we have used or cited take slightly different approaches to definitions and scope, making it difficult for us to keep our language consistent. Target 3 itself also refers to both Indigenous and traditional territories and Indigenous Peoples and local communities. When referring to people who reside within ITTs we now consistently use the term “Indigenous Peoples and custodian communities”. This has resulted in the following edits:  - Line 182 - Line 629

	 - Line 693 - Line 734 - Lines 205 and 217 (Supplementary Information) The exception is where the reference is in the context of another study which is about Indigenous Peoples and Local Communities (such as the WWF 2021 report, citation number 22).  - At lines 107-111 we have split a sentence into two to avoid implying that the WWF report gives a measure of Indigenous and Traditional Territories. “The appropriate recognition of ITTs could be key to achieving Target 3, given their crucial role in protecting biodiversity and in environmental governance. For example, estimates suggest that land owned or governed by Indigenous Peoples and local communities covers at least 32% of Earth's terrestrial surface area, and likely more due to a lack of data and under reporting²².”
Finally, Figures 2 and 3 are wonderfully rich. Consider including a note to reiterate that the “wild harvesting” data apply only to tropical regions.	We thank the reviewer for highlighting this. As noted in our response to a previous comment on broader acknowledgement, we have clarified in the caption of Figure 2 that the wild harvesting data are tropical. The caption now reads: “Livelihoods of resident population of expansion areas, expressed as the percentage of population by (C1) wild harvesting status of households (tropical population)...”. Figure 3 does not include wild harvesting, so no changes were necessary to its caption.

We hope that the revisions meet your expectations and those of the reviewers. We would like to thank you again for your time and consideration.

Sincerely,

Javier Fajardo & Chris Sandbrook

(on behalf of all co-authors)